# Impact of paleoclimate on present and future evolution of the Greenland Ice Sheet

**Hu Yang**[1]*, **Uta Krebs-Kanzow**[1], **Thomas Kleiner**[1], **Dmitry Sidorenko**[1], **Christian Bernd Rodehacke**[1,2], **Xiaoxu Shi**[1], **Paul Gierz**[1], **Lu Niu**[1], **Evan J. Gowan**[1,4], **Sebastian Hinck**[1], **Xingxing Liu**[1,3], **Lennert B. Stap**[1,5], **Gerrit Lohmann**[1]

1 Alfred Wegener Institute Helmholtz Centre for Polar and Marine Research, Bremerhaven, Germany, 2 Danish Meteorological Institute, Copenhagen, Denmark, 3 State Key Laboratory of Loess and Quaternary Geology, Institute of Earth Environment, Chinese Academy of Sciences, Xi'an, China, 4 Department of Earth and Environmental Sciences, Kumamoto University, Kumamoto, Japan, 5 Institute for Marine and Atmospheric Research Utrecht, Utrecht University, Utrecht, Netherlands

* hu.yang@awi.de

**Data Availability Statement:** All relevant data are within the paper and its Supporting information files.

## Abstract

Using transient climate forcing based on simulations from the Alfred Wegener Institute Earth System Model (AWI-ESM), we simulate the evolution of the Greenland Ice Sheet (GrIS) from the last interglacial (125 ka, kiloyear before present) to 2100 AD with the Parallel Ice Sheet Model (PISM). The impact of paleoclimate, especially Holocene climate, on the present and future evolution of the GrIS is explored. Our simulations of the past show close agreement with reconstructions with respect to the recent timing of the peaks in ice volume and the climate of Greenland. The maximum and minimum ice volume at around 18–17 ka and 6–5 ka lag the respective extremes in climate by several thousand years, implying that the ice volume response of the GrIS strongly lags climatic changes. Given that Greenland's climate was getting colder from the Holocene Thermal Maximum (i.e., 8 ka) to the Pre-Industrial era, our simulation implies that the GrIS experienced growth from the mid-Holocene to the industrial era. Due to this background trend, the GrIS still gains mass until the second half of the 20th century, even though anthropogenic warming begins around 1850 AD. This is also in agreement with observational evidence showing mass loss of the GrIS does not begin earlier than the late 20th century. Our results highlight that the present evolution of the GrIS is not only controlled by the recent climate changes, but is also affected by paleoclimate, especially the relatively warm Holocene climate. We propose that the GrIS was not in equilibrium throughout the entire Holocene and that the slow response to Holocene climate needs to be represented in ice sheet simulations in order to predict ice mass loss, and therefore sea level rise, accurately.

## Introduction

Under global warming, melting of the Greenland Ice Sheet (GrIS) is expected to be one of the dominant contributors to future sea level rise [1, 2]. However, uncertainties remain regarding

**Funding:** This work was supported through grant (Global sea level change since the Mid Holocene: Background trends and climate-ice sheet feedbacks) from the Deutsche Forschungsgemeinschaft (DFG) as part of the Special Priority Program (SPP)-1889 'Regional Sea Level Change and Society' (SeaLevel). C. Rodehacke has been financed through the German Federal Ministry of Education and Research (Bundesministerium fur Bildung und Forschung: BMBF) project ZUWEISS (grant agreement 01LS1612A) and through the National Centre for Climate Research (NCFK, Nationalt Center for Klimaforskning) provided by the Danish State. H.Y., S.X. and X.L are partly funded by the open fund of State Key Laboratory of Loess and Quaternary Geology, Institute of Earth Environment, CAS (SKLLQG1920). Development of PISM is supported by NASA grant NNX17AG65G and NSF grants PLR-1603799 and PLR-1644277.

**Competing interests:** NO authors have competing interests.

how much climate change will affect the speed and amount of GrIS retreat. Systematic observations cover only a few decades and cannot inform about the long-term response of the GrIS to the ongoing, intensifying warming. The past and future evolution of the GrIS, as either reconstructed or simulated, provides an important insight into the processes which become important on longer time scales.

Ice sheet reconstructions based on indicators of past ice extent and relative sea level data indicate that the GrIS was substantially larger during the onset of the last deglaciation (18–16 kiloyear before present, hereafter referred to as ka) than present [3–6]. In contrast, the GrIS reached a minimum extent at around 6–4 ka after a period of warm climate called the Holocene Thermal Maximum (8 ka) [5, 7–15]. Besides these direct geologically based reconstructions, several modelling efforts on the paleo GrIS have been conducted by using forcing constrained by paleo proxies of temperature, ice extent and relative sea level [16–19]. For instance, several studies [16, 17] simulated the GrIS since the Last Glacial Maximum (LGM). Their results suggest that the GrIS reached a maximum size around 16.5 ka, corresponding to an additional ice volume of 4.6 m to 5.1 m sea level equivalent (SLE) relative to present-day, while a minimum size was approached around 5–4 ka with ice volume being reduced by 0.16 m to 0.17 m SLE relative to present-day. Another extended GrIS simulation [20] covering the past two glacial-interglacial cycles proposed that the GrIS contributed 1.46 m and -2.59 m to global sea level during the last interglacial and LGM, respectively. Under an idealised forcing by prescribing temperature and precipitation following several Greenland ice core records, Nielsen and his colleagues simulated the evolution of the GrIS during the last ten thousand years [18]. Their simulations suggest that the Holocene Thermal Maximum potentially reduced the GrIS to a minimum size of 0.15 m to 1.2 m SLE smaller than present at around 9–7 ka. By forcing the GrIS using improved seasonal temperature reconstruction, another simulation [21] obtained a Holocene minimum in GrIS mass of 0.55 m SLE below present day.

Besides the studies on the past evolution of the GrIS, a great effort has been carried out to predict the future evolution of the GrIS [22–30]. Depending on the greenhouse gases emissions scenarios and numerical model strategies, studies have projected that the GrIS may contribute to 2 cm to 33 cm of sea level rise by the end of 21st century. By integrating the GrIS model over the next millennium, Aschwanden et al [31] illustrated that the GrIS may melt away under the strongest warming scenario.

Many studies find that different initialisation methods affect the future projections of the GrIS [23, 24, 32, 33]. Sensitivity simulations [32] indicate that the short-term GrIS evolution projected by an ice-sheet model is strongly influenced by the initial conditions set in the simulation. Therefore, it is essential for numerical simulations to provide an initial model state that is consistent with the climate forcing [24]. Previously, two major initialisation strategies were widely used, i.e., paleo-spinups and equilibrium-spinups. The paleo-spinup strategy uses the evolution of past climate to force the ice sheet model to present, while the equilibrium-spinup applies observed climate from the present (or pre-industrial) to force the model into an equilibrium state. Previous literature [23] noticed that simulations with paleo-spinup produce results closer to observations than those with equilibrium-spinup.

In this study, we simulate the evolution of the GrIS from the last interglacial to the year 2100 using forcing prescribed by a climate model. The impact of paleoclimate, especially the relatively warm Holocene climate, on present and future evolution of the GrIS is explored by comparing sensitivity simulations that do not contain information of paleoclimate change.

## Experiment design

### Model setup

The evolution of the GrIS from the last interglacial (125 ka) to 2100 AD is simulated by the Parallel Ice Sheet Model (PISM, version 0.7.3, [34, 35]) with a resolution of 5 km. We employ PISM with a hybrid stress balance of non-sliding shallow ice approximation and the shallow-shelf approximation (SIA+SSA). The ice velocity is determined by a linear superposition of the SIA and the SSA velocities. In areas where the basal sliding is negligible, the SIA dominates the ice velocity, while for the ice shelves and fast flowing regions, the ice flow is primarily controlled by the SSA. We use the Lingle and Clark bed deformation model [36, 37] to simulate the solid earth deformation. By doing so, the deformation of the Earth's crust and the related regional relative sea level variations are taken into account. The Positive Degree Day (PDD) scheme is used to compute the ice ablation [38]. A constant temperature lapse rate of 5˚C km$^{-1}$ is adopted to account the elevation-induced changes in the near-surface temperature, which also impacts the surface mass balance. The eigen-calving and thickness calving parameterizations are used to determine the calving-front dynamics. Eigen-calving is a simple formula for first-order kinematic contribution to iceberg calving, in which volume loss through calving at the ice front is proportional to the determinant of the strain rate tensor, i.e. the product of its eigenvalues [39]. The thickness calving scheme is a simplified parameterization which removes the ice shelf when its thickness exceeds a certain thickness threshold (i.e., thickness_calving_threshold in PISM). Calving takes place when the ice front meets the threshold from either of these two calving laws.

Considering that the GrIS covered a large portion of the continental shelf of Greenland during the last glacial period [3, 4], the chosen model domain is selected to cover it entirely (as shown in Fig 1). A global sea level reconstruction [40] and basal geothermal heat flux [41] are used as external forcing. These boundary conditions and all other input data are bilinearly interpolated onto the 5 km model grid. The present-day bedrock elevation and ice thickness from the ETOPO1 (Fig 1, [42]) are used as the initial conditions at the very beginning of our simulation, i.e., the last interglacial (125 ka).

Previous modelling studies of the GrIS, such as [16, 17], use geological data to tune the ice extent and thickness. In these simulations, the ice extent is set to follow these observations by

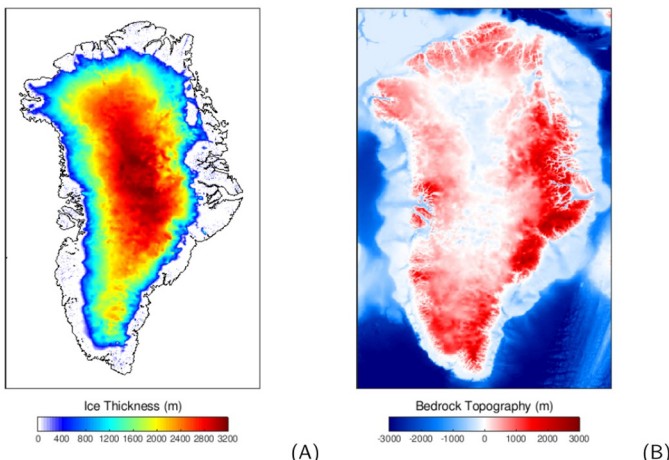

**Fig 1. The model domain of the ice sheet model.** The model domain of the ice sheet model. The left subpanel shows the ice thickness used to initialise the model (shading) and the contemporary coast line (black contour line) while the right subpanel depicts the corresponding bedrock topography. These data are derived from the ETOPO1 data set [42].

adjusting the grounding line parameterizations and ice extent, but this may cause violations in the ice dynamics of the GrIS. To prevent the ramifications of this kind of tuning of the ice dynamics, we do not artificially constrain the ice extent or thickness, except to approximate present day conditions. The ice sheet develops freely, which is driven solely by the changing climate forcing. We tuned the enhancement factor (sia_e = 5) to get a reasonable ice volume (i.e., close to 7.4 m SLE) for the Pre-Industrial era (i.e., the end of the paleo GrIS simulation).

It should be noted that most of our parameterizations are similar to a previous study [43]. However, we do not downscale the precipitation with respect to elevation changes, because compared to [43], the climate forcing derived from the Alfred Wegener Institute Earth System Model (AWI-ESM) has a relatively higher spatial resolution that resolves a precipitation pattern that resembles observations (Section: Features of climate forcing and S1 Fig in S1 File).

### Transient climate forcing strategies

The climate forcing for the GrIS comes from the AWI-ESM [44, 45]. The AWI-ESM consists of the atmosphere and land-surface model ECHAM6.3 [46] coupled with the Finite Element Sea ice-Ocean Model (FESOM 1.4, [47]). The simulations performed in this study used the AWI-ESM with T63 spectral resolution in the atmosphere ($\sim 80$ km over the GrIS). Ocean and sea ice were simulated on a mesh with resolution varying from nominal one degree in the interior of the ocean to $1/3°$ in the equatorial belt and $\sim 24$ km north of 50°N. This configuration of AWI-ESM is rather costly. Hence, it is unrealistic to simulate the entire paleoclimate history to construct a continuous forcing for the simulations of the GrIS. Therefore, we perform six time-slice simulations representing different climate stages (i.e., 127 ka, 21 ka, 9 ka, 8 ka, 6 ka, Pre-Industrial) following the protocol of the Paleoclimate Modelling Intercomparison Project phase 4 (PMIP4, [48, 49]). This protocol describes the modifications of the orbital parameters, concentration of greenhouse gases, topography and ice sheet coverage to simulate past climate. Note that some of the above simulations were created as part of the fourth phase of the Paleoclimate Modelling Intercomparison Project (PMIP4). Detailed information of these simulations can also be found in the online Coupled Model Intercomparison Project Phase 6 (CMIP6) documentation [50–53]. The model intercomparison indicates that AWI-ESM produces reasonable paleoclimate patterns when compared with other PMIP4 models [54–60].

The time-slice climate conditions are interpolated temporarily to obtain a continuous transient paleoclimate forcing. As in [43], the glacial index method [61] is used to prescribe the climate forcing evolution for the simulation from the last interglacial to the Pre-Industrial era. Under such an approach, the atmospheric forcing (i.e., precipitation and near-surface air temperature) from the last interglacial (127 ka) to the LGM (21 ka) is generated as a linear combination between the two corresponding climatic conditions obtained from the AWI-ESM. Here, the NGRIP ice core record sampled from the summit of the GrIS [62] is used as an index to derive the temporal evolution of climate from the last interglacial to the LGM. The glacial index method is widely used to generate transient climate forcing for the paleo ice sheet modelling [43, 63–67].

During the last deglaciation, climate changes, especially those associated with the retreat of the GrIS, are primarily driven by variations of Northern Hemisphere summer temperature. The oxygen isotope signal in the NGRIP ice core, however, is strongly affected by the fluctuation of the Atlantic meridional overturning circulation [68], which is predominantly a winter temperature anomaly [21]. To avoid overestimating summer temperature variations from 21 ka to Pre-Industrial era, the AWI-ESM time-slice simulations of 21 ka, 9 ka, 8 ka, 6 ka, and Pre-Industrial are linearly interpolated to derive a continuous climate forcing. Such linear

transient strategy also lies in the fact that we have better temporal coverage of climate simulations after 21 ka, in comparison with the climate snapshots available before 21 ka.

Prior investigations [19, 69–74] revealed that oceanic temperature is an important condition for the evolution of the GrIS, especially during the last deglaciation. To take into account the effects of the ocean, the Greenland weighted-average coastal ocean temperature (upper 100 meters) is used as an oceanic temperature forcing. Unlike the prescribed atmospheric forcing that contains spatial and seasonal variations, we apply a simplified, spatially uniform ocean temperature forcing using the glacial index method, based on the yearly mean ocean temperature between the LGM (−1.43˚C) and the Pre-Industrial (−0.19˚C) experiments. The ice shelf–ocean interaction scheme introduced by Beckmann and Goosse [75] is applied to the ocean component with a melt factor of 0.01. The sub-shelf ice temperature is set to the pressure melting point and the sub-shelf melt rate is assumed to be proportional to the heat flux from the ocean into the ice.

In order to prescribe the climate forcing for both present and future projections of the GrIS, a transient climate simulation covering the period of 1850–2100 is carried out by the AWI-ESM. This transient climate simulation consists of two parts (i.e., historical transient (1850–2005) and Representative Concentration Pathway 8.5 (RCP8.5) future projection (2006–2100)), following the protocol of the Coupled Model Intercomparison Project, Phase 5 [76]. The 250 years of monthly mean near-surface temperature, precipitation and yearly mean ocean temperature are then used to force our industrial era GrIS simulation between 1850 and 2100.

## Ensemble simulations

In addition to the above described GrIS simulation, we perform another five simulations by changing the PISM parameters on sliding (till_effective_fraction_overburden), calving (thickness_calving_threshold), lapse rate (temp_lapse_rate), sub-shelf melting (meltfactor_pik) and ice flow (sia_e) (see Table 1). These parameters are selected because their actual ranges have large uncertainties. The parameter till_effective_fraction_overburden affects the effective pressure in the till. A lower till_effective_fraction_overburden value helps the ice to slide more easily. The parameter thickness_calving_threshold provides the thickness threshold for ice shelf calving. The temp_lapse_rate gives the lapse rate for vertical air temperature downscaling. The meltfactor_pik determines the efficiency of the heat exchange between the ocean water and marine terminating ice. Higher meltfactor_pik promotes more ocean-ice heat exchange. The flow enhancement factor for the shallow ice approximation is given by the parameter sia_e. More detailed information on these parameters can be found in PISM user's manual [34, 35]. Owing to the gravitational effects of the Laurentide Ice Sheet [6], the magnitude of sea level change around the GrIS is likely smaller than the scaled global mean sea level based on benthic

**Table 1. List of ensemble simulations in this study.**

| Exp Name | Parameter description | Parameter name | Sensitivity simulation | Default simulation |
|---|---|---|---|---|
| **Sliding** | basal sliding | till_effective_fraction_overburden | 0.0096 | 0.01 |
| **Calving** | thickness calving | thickness_calving_threshold | 250 | 200 |
| **LapseR** | lapse rate | temp_lapse_rate | 6 | 5 |
| **MeltF** | ocean melt factor | meltfactor_pik | 0.009 | 0.01 |
| **SIA** | shallow ice approximation | sia_e | 4 | 5 |
| **SeaL** | sea level forcing | sea level scale | 0.9 | 1 |

$\delta$18O values [77]. Considering the uncertainties on regional sea level, we perform a simulation by scaling the forcing of global mean sea level [40] with a factor of 0.9.

## Features of climate forcing

Figs 2 and 3 show the 100 year mean climate patterns from the time slice experiments and the final 10 years (i.e., 2091–2100) of the transient simulation as anomalies relative to Pre-Industrial conditions. Since the climate primarily influences the GrIS by the year-round accumulation of snow and ablation during the summer melt season, we present the summer (JJA) temperatures and yearly mean precipitation for comparison.

The spatial pattern of Pre-Industrial summer temperature primarily reflects Greenland's topography, with temperatures falling well below −20˚C around the summit of the GrIS. Temperatures near or above melting point are found around the margins. Over the ice sheet, the highest temperatures prevail in the ablation zones of southwest Greenland (Fig 2) while

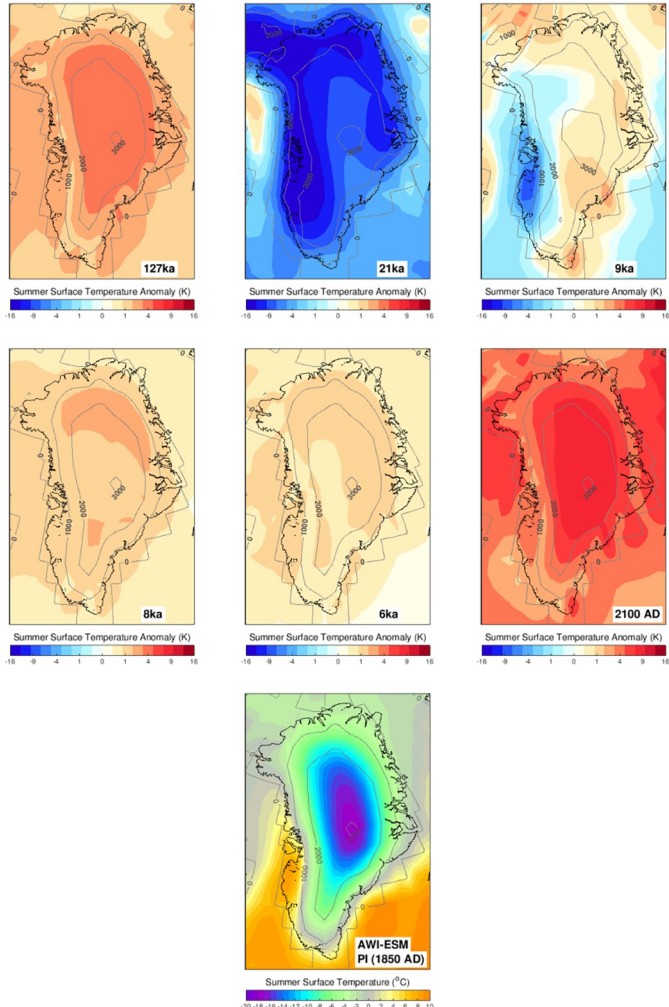

**Fig 2. The AWI-ESM prescribed summer (June-July-August: JJA) 2m-air temperature in different periods (plotted as anomaly to the Pre-Industrial conditions).** The absolute 2m-air temperature at the Pre-Industrial era is presented in the bottom panel. The contemporary coast line is highlighted in the black line (see Fig 1 for details). The grey contours give the ice sheet geometry used in the climate model.

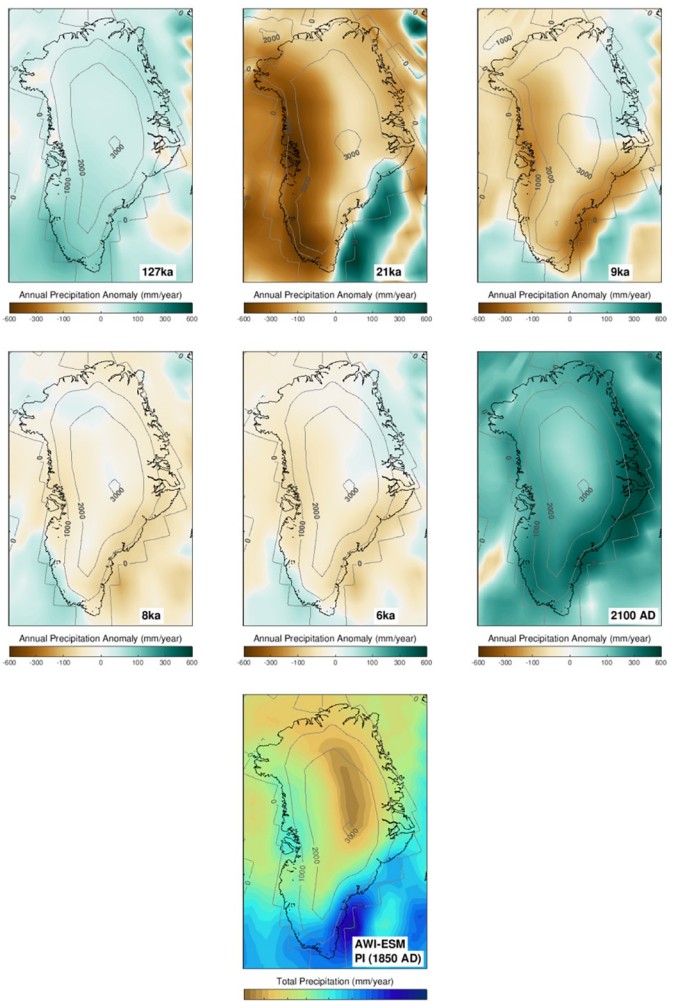

**Fig 3. Same as Fig 2, but for annual mean precipitation.**

temperature of the same elevation range are generally lower in the northeast due to the uneven distribution of solar radiation and the effects of the sea ice covered Arctic Ocean (not shown). Compared to the Pre-Industrial summer temperature, the last interglacial period exhibits about ≈4˚C higher near-surface air temperature. Temperature anomalies are mostly uniform and follow the large-scale glacial-interglacial climate evolution caused by changing summer insolation [78] and greenhouse gas concentrations. Compared to the Pre-Industrial, we find that the coldest conditions in the 21 ka experiment are about 13˚C to 14˚C colder than the Pre-Industrial conditions. There are spatially heterogeneous temperature anomalies at 9 ka showing that western Greenland is colder than Pre-Industrial levels, while the rest of the ice sheet is warmer. The snapshot experiments of 8 ka and 6 ka are 1˚C to 2˚C warmer than Pre-industrial levels. Our climate model simulations well capture the relatively warm Holocene climate as indicated by multiple proxy records [21, 79–83]. From the Pre-Industrial (1850 AD) onward, Greenland summer temperatures rise again at an accelerating rate due to increasing concentrations of greenhouse gases. Under the RCP8.5 scenario, the simulated climate around 2100 over Greenland has warmed by about 6˚C relative to Pre-Industrial.

In addition to the changing climate, the spatial temperature anomaly over the GrIS is also influenced by the temporal changes in the ice sheet geometry. More specifically, we have applied a different topographic boundary conditions in the simulations of the 21 ka and 9 ka experiments, while the modern topography is used in the 127 ka, 8 ka, 6 ka, Pre-Industrial and 1850–2100 transient experiments. Compared to the Pre-Industrial, the 21 ka experiment is based on a more extensive ice sheet. Accordingly, the climate of Greenland exhibits pronounced cooling around the margins and only moderate cooling near the modern summit. The 9 ka experiment shows colder conditions in western Greenland, due to the cooling effect caused by the residual Laurentide Ice Sheet (not shown) and differences in local topography. To eliminate the impact from the ice sheet geometry, the temperature anomalies at the 700 hPa tropospheric level are also given in S2 Fig in S1 File. We note that temperature changes with similar amplitude are still visible at the troposphere level above the ice sheet, implying that the large-scale climate changes are the dominant contributors for the simulated temperature anomalies.

Similar to the present-day observed precipitation pattern [84, 85], the annual mean precipitation of our Pre-Industrial experiment exhibits a pronounced maximum along the southeast portion of the GrIS and low accumulation at high altitudes (Fig 3 and S1 Fig in S1 File). Due to the cold climate of the LGM, the 21 ka experiment has the lowest accumulation rates over most parts of the ice sheet. Precipitation intensifies after the LGM. The 9 ka climate is still considerably drier than Pre-Industrial while both the 8 ka and 6 ka climate states almost reach Pre-Industrial accumulation rates. In general, we find the lowest precipitation rates in the coldest climate (i.e., LGM), and the highest precipitation rate in the warmest climate (i.e., 2100 AD in RCP8.5 experiment). During the early and mid-Holocene (i.e., 9 ka, 8 ka, 6 ka), when the climate conditions were not significantly different from the Pre-Industrial, the precipitation anomaly is relatively heterogeneous.

## Past evolution of the GrIS with a focus on the Holocene

Fig 4 provides the evolution of the prescribed climate forcing and ice volume of the GrIS from the last interglacial to the Pre-Industrial era. In general, the climate gets colder from the last interglacial through the last glacial period, contributing to increasing the ice volume of the GrIS. The climate warms from the LGM to the Holocene, which causes the the GrIS to retreat. This suggests that the temperature, rather than the snowfall rate, is the primary driver for the evolution of the GrIS. Focusing on the periods of extremes in Greenland's climate and ice volume, we notice that the recent coldest climate is around 26 ka, followed by a warming afterwards. Despite the warming trend, the GrIS still grows for another nine thousand years, and reaches its maximum volume at around 18–17 ka. The Greenland temperature peaks at around 8 ka while the minimum ice volume is obtained 2–3 thousand years later, at around 6–5 ka. Such a delayed ice volume response to climate change happened in all of the sensitivity simulations (Fig 4), independent of the changes in model parameters.

Given that the simulated evolution of the GrIS before the Holocene has large uncertainties in terms of initial conditions, sea level forcing and the ocean temperature forcing, we primarily focus our analysis on the results from the Holocene Thermal Maximum (8 ka) onward. During the Holocene, the GrIS is mostly terrestrially terminating (S3 Fig in S1 File, and [86]), suggesting that the atmospheric forcing is the primary driver for the GrIS. As shown in Fig 5, both summer insolation [78] and the Greenland ice core proxy records [62] reveal a cooling trend from the Holocene Thermal Maximum (8 ka) to the Pre-Industrial era, in agreement with our model simulations. Despite the cooling trend starting from 8 ka, the GrIS continues to lose mass until 6–5 ka. The dynamic ice loss slows down and reaches a balance with the surface

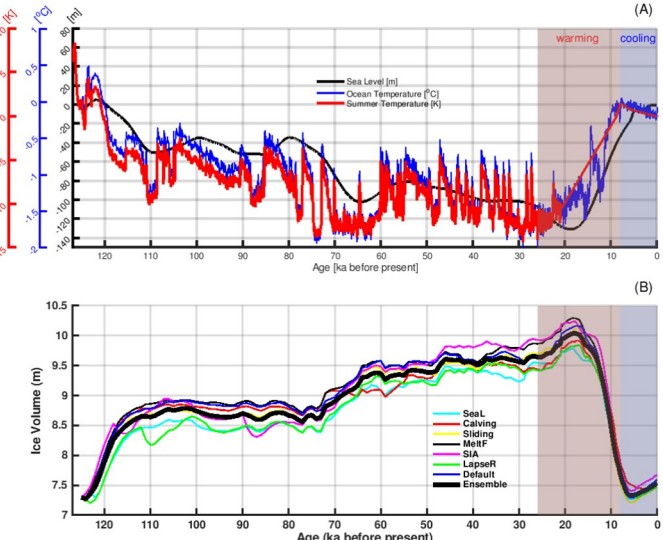

**Fig 4. The evolution of Greenland's climate and simulated ice volume of the GrIS from the last interglacial to the Pre-Industrial era.** (A): The prescribed forcing of summer (JJA) atmospheric temperature (red line), yearly mean ocean temperature (blue line) and global mean sea level (black line). (B): The simulated ice volume evolution of the GrIS. The coloured lines are the results from individual ensemble members (see Table 1). The thick black line gives the ensemble mean values. The red shadow area marks the time period when Greenland's climate gets warmer, and the blue patch illustrates the time when Greenland's climate colds.

mass balance around 6–5 ka when the GrIS reaches its minimum size, with an ice volume of 0.20 m SLE smaller than the Pre-Industrial volume. The minimum volume of the GrIS also coincides with the smallest ice extent (S4 Fig in S1 File). After 5 ka, summer cooling and the associated increasing surface mass balance lead to a re-advance of the GrIS, causing expanding ice extent and increasing ice volume. Approaching to the Pre-Industrial era, the surface mass balance is higher than the rate of dynamic ice loss, denoting a background growth of the GrIS. Our simulations imply that the GrIS was not in an equilibrium state at any time during the entire Holocene. There is a background trend of GrIS growth when approaching the Pre-Industrial era.

## Impact of Holocene climate on the present and future evolution of the GrIS

The above results imply that the ice volume response of the GrIS strongly lags climate changes. Therefore, the current evolution of the GrIS is not only controlled by the present climate change, but is also affected by the climate of the past. In order to investigate the actual impact of paleoclimate on the present and future evolution of the GrIS, we continue our paleo GrIS simulation into the year 2100 under the forcing of an industrial era climate scenario (1850–2100). As shown in Fig 6a, under the forcing of increasing greenhouse gases (black line), the mean Greenland-wide summer temperature rises steadily after the 1900s, and accelerates after the 1970s (red line). Despite the forcing of a warming climate, we still observe growth of the GrIS from the 1850s to the 1970s (Fig 6b, black line). This is primarily because the GrIS is still growing due to the cooling from the mid to late-Holocene. Such growth is reversed around the 1970s when Greenland's surface temperature experiences a prominent warming. Our simulations show that the mass loss of the GrIS starts around the 1970s and keeps accelerating afterwards due to enhanced anthropogenic warming. Based on the RCP8.5 scenario, the simulated

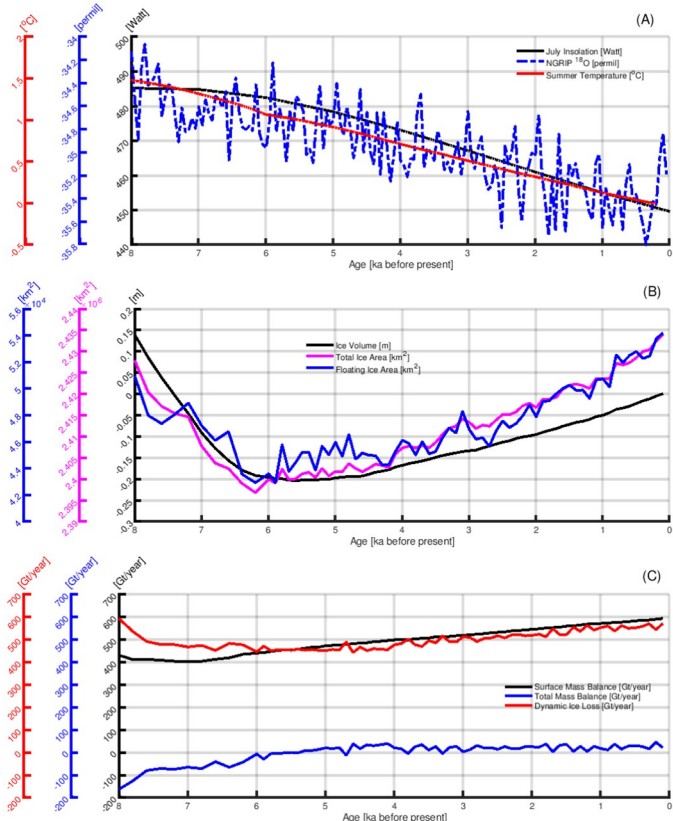

**Fig 5. Evolution of Greenland Ice Sheet from the Holocene Thermal Maximum (i.e., 8 ka) to the Pre-Industrial era.** A, top: Blue line: the oxygen $\delta^{18}O$ isotope record from the NGRIP ice core [62]. Red line: the prescribed forcing of summer (JJA) temperature. Black line: July insolation at 70˚N [78]. B, center: Black line: ice volume of the GrIS. Pink line: total ice area of the GrIS. Blue line: floating ice area. C, bottom: Black line: surface mass balance of the GrIS. Blue line: total mass balance, positive indicate mass gain, and vice versa. Red line: rate of dynamic ice loss, i.e., discharge ice flux. As the sub-shelf ice flux and grounded basal mass flux are one order of magnitude smaller than that of the surface mass balance and the dynamic ice loss, the total mass balance of the GrIS is primarily controlled by the surface mass balance and dynamic ice loss. The above results are all based on the ensemble mean of the seven ensemble simulations.

Greenland-wide summer surface temperature increases about 6˚C by the year 2100. The warming promotes the disintegration of the GrIS which raises the sea level by 4.5 cm SLE by 2100. Given that the RCP8.5 warming (around 6˚C, Fig 2) by 2100 is much stronger than the warming during the mid-Holocene (around 1 degree Celsius, Fig 2), and the 4.5 cm SLE melting of GrIS by 2100 is very likely to be a prelude to further melting. Our preliminary simulation indicates that under the steady climate forcing taken from the RCP8.5 climate state by the year 2091–2100, the entire GrIS will melt within a few millennia (not shown).

To explore how the GrIS evolves without the impact of Holocene climate change, we conduct another simulation forced by the identical industrial climate (1850–2100) as the above simulation, but initialised with a different strategy. In this simulation, the initial condition is obtained by continuing the paleo GrIS simulation for ten thousand years under the forcing of the Pre-Industrial steady state climate. As described previously (Section: Past evolution of the GrIS with a focus on the Holocene), the GrIS was not in an equilibrium state during the entire Holocene. Approaching the Pre-Industrial era, the GrIS was still adjusting to a relatively cool Pre-Industrial climate in comparison to the previous warmer Holocene climate. Under steady Pre-Industrial climate forcing, the GrIS grows from an ice volume of ∼7.54 m SLE to a

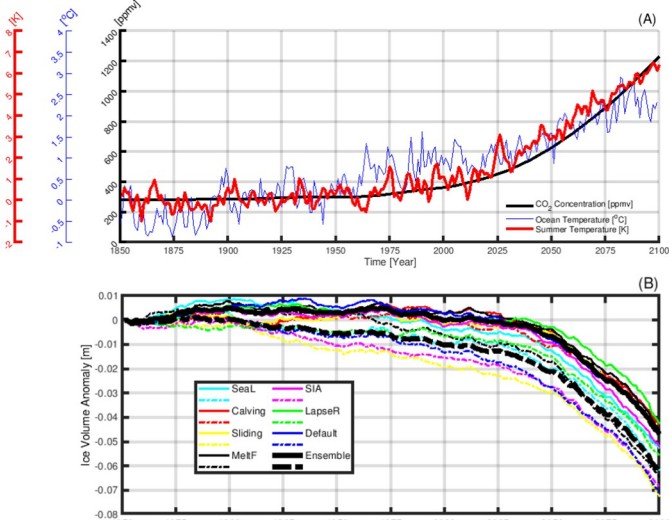

**Fig 6. The evolution of Greenland's climate and ice volume during 1850–2100.** (A): Black line: concentration of
$CO_2$ forcing in our climate model simulation. Blue line: yearly mean ocean temperature forcing. Red line: Greenland-
wide summer temperature anomalies in respect to the Pre-Industrial values (100 years of climatology mean). (B):
Anomaly of ice volume of the GrIS with respect to the Pre-Industrial values. The solid lines are the simulations using
initial conditions obtained by the paleo-spinup, while the dashed lines are the results using initial conditions obtained
by a ten thousand year Pre-Industrial equilibrium-spinup. The coloured lines are the results from individual ensemble
members (see Table 1), and the thick black lines are the results of the ensemble mean.

quasi-equilibrium state, holding ∼0.10 m SLE more ice after ten thousand years of equilib-
rium-spinup (Fig 7a). Increased ice volume is primarily found over the northeast (Fig 8a),
where the precipitation rate is relatively low. Since drier areas tend to need more time to
respond to a cooling climate, a long relaxation under PI (cold) conditions is critical for the
northeast to gain mass. The ice thickness increase modelled at the northeast, northwest and
west is additionally sustained by the creation of ice shelves that, by exerting resistance into the
interior of the ice sheet, prevent ice-mass loss. This buttressing effect is well seen by the slow-
down in the velocity modelled in these regions (Fig 6b).

Compared to the simulation initialised from the paleo-spinup, the equilibrium-spinup
shows almost immediate ice loss after the 1890s, almost one century earlier than the simula-
tions starting from the paleo-spinup (Fig 6). Moreover, the estimated contribution of sea level
rise from melting the GrIS is on the order of 6.2 cm SLE by 2100, which is larger than the
results from the simulation initialised from the paleo-spinup (black line in Fig 6). Our results
indicate that if the late-Holocene background trend of growth of the GrIS is not included, the
onset and magnitude of mass loss (and therefore sea level rise) of the GrIS can both be overes-
timated by an ice sheet model.

To understand the behaviour of the GrIS initialised from different initial conditions, we
evaluate the discrepancies in the ice temperature and dynamic ice loss between those two ini-
tial condition setups. As shown in Fig 7b, when approaching the Pre-Industrial era, the ice
temperature of the GrIS is still increasing, owing to the relatively cold climate. Overall, the
equilibrium condition has an 0.4˚C higher ice temperature than the condition obtained from
paleo-spinup. Moreover, at the end of the equilibrium-spinup, the surface mass balance of the
GrIS reaches a quasi-balance with the dynamic ice loss, which is stronger than the paleo-
spinup (Fig 7c). The higher dynamic ice loss rate is also expressed by an overall higher surface
ice velocity (Fig 8b). Due to these differences in ice temperature and rate of dynamic ice loss,

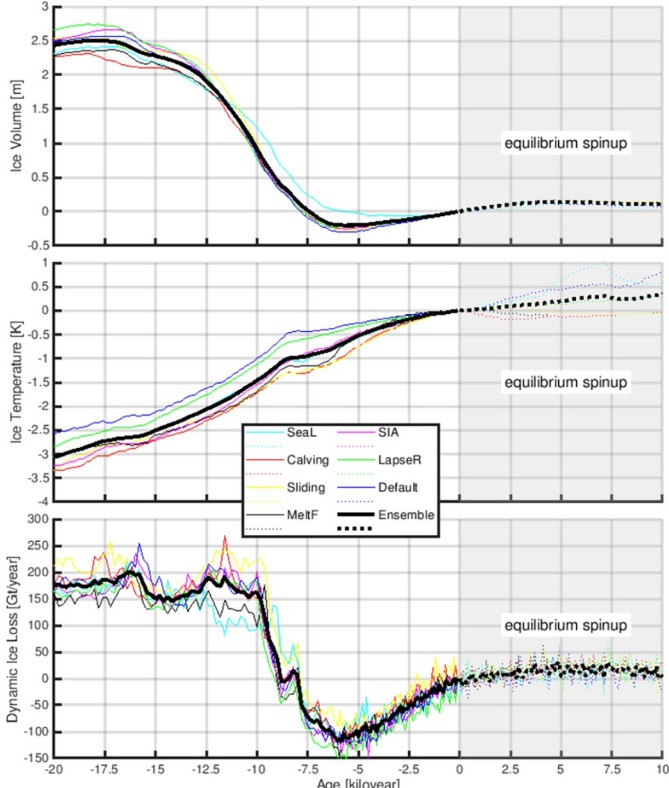

**Fig 7. The simulated evolution of the (A, top) ice volume, (B, center) ice temperature (average over entire GrIS) and (C, bottom) dynamic ice loss in the paleo-spinup (solid lines) and the equilibrium-spinup (dashed lines).** All results are given as anomalies with respect to the corresponding Pre-Industrial (i.e., 1850) values. The coloured lines are results from individual ensemble members, and the thick black lines are the ensemble mean.

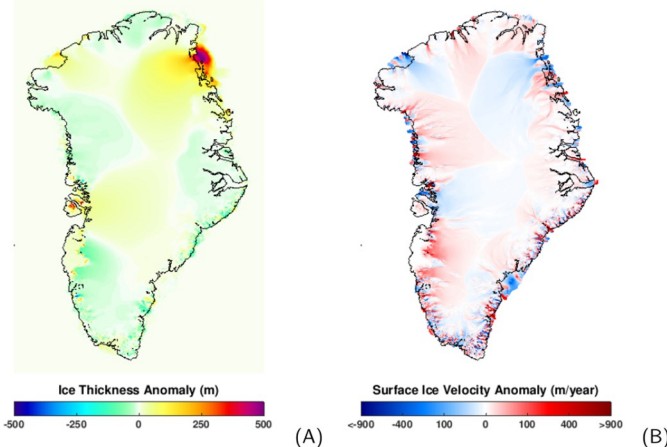

**Fig 8. (A, left) Ice thickness and (B, right) surface ice velocity anomaly between the initial conditions obtained from the equilibrium-spinup and paleo-spinup (equilibrium minus paleo).** Black line represents the coastline. As shown, individual glaciers have distinct responses. Overall, northeastern Greenland contributes to a larger GrIS ice volume in comparison with the that obtained from the paleo-spinup. With respect to the surface ice velocity, large portions of the GrIS show faster ice velocity in the equilibrium-spinup with respect to that obtained from paleo-spinup. Results are the ensemble mean of the seven ensemble members.

the GrIS simulation with the equilibrium-spinup can overestimate the onset and magnitude of the melt of the GrIS under the forcing of anthropogenic warming.

## Discussion

Reconstructing the past evolution of the GrIS is challenging owing to the lack of observations both on temporal and spatial scales. By constraining the ice sheet model using field observations of relative sea level and ice extent, previous studies [5, 16, 17] propose that the GrIS reached its maximum size around 18–16 ka. Paleo proxies of temperature, ice extent and ice elevation [8, 11, 81–83, 87] combined with ice sheet model simulations [16–18] reveal that the GrIS retreated to a minimum size most likely between 7–4 ka.

In the present study, we revisit the past evolution of the GrIS using ice sheet model (i.e., PISM). Our simulations reproduce the timing of both maximum (around 18–17 ka) and minimum (around 6–5 ka) sizes of the GrIS, which are close to the previous reconstructions. We find that the times when the GrIS approached to its maximum and minimum size are several thousand years delayed compared to the times when the Greenland climate reaches its coldest (around 26 ka) and warmest (around 8 ka) conditions. Without any constraints imposed from observations, our simulations illustrate that the maximum size of the GrIS (at around 18–17 ka) is about 2.51 m SLE bigger than today, and the minimum size is around 0.20 m SLE smaller than today during the mid-Holocene. Previous GrIS simulations constrained by observations suggest that during the last deglaciation, the GrIS may hold 4.6 m to 5.1 m SLE more ice relative to present-day [16, 17]. Our simulated magnitudes show discrepancies with the previous studies, likely because the simulations have not been constrained by any observation. In addition, a simplified glacial index method could also contribute to such a discrepancy. We realize that the simulated ice volume of the GrIS varies depending on the choice of model parameters and also the prescribed climate forcing. Temperature reconstruction from boreholes [79], ice cores [21, 80, 81] and lake sediments [82, 83] indicate that the regional Greenland peak summer temperature during the Holocene Thermal Maximum was 2°C to 7°C warmer than the Pre-Industrial era. Our model simulation tends to underestimate this warming with a magnitude of around 1°C. Therefore, the simulated minimum ice volume of the GrIS during the mid-Holocene may be underestimated. With forcing prescribed by ice core proxies, two previous studies [18, 21] obtained a much larger retreat of the GrIS, which contributes to a sea level rise of about 0.15–1.2 and 0.55 m during the mid-Holocene, respectively. Furthermore, we can assume that the response of the GrIS to Holocene climate change is additionally underestimated in our experiments, as the applied PDD scheme does not account for the variations in insolation on orbital time scales. During the last interglacial the change in insolation substantially contributed to surface melt on the GrIS [88, 89]. For the mid-Holocene, estimation according to [90] suggests that the PPD schemes will underestimate the surface melt by approximately 10% due to the neglected effect of insolation. Disregarding quantitative differences in ice volumes at different periods, our simulations and previous studies consistently reveal that the ice volume response of the GrIS lags climate change by millennia, independently of the setup of the ice sheet model and climate forcing.

Our GrIS simulations use an offline prescribed climate to force the GrIS, in which several processes are simplified. First, the paleoclimate evolution is prescribed using an idealised glacial index method, which is unable to account for the spatial inhomogeneity of climate variability. Second, a spatially uniform ocean temperature is adopted to force the GrIS. This neglects the role of regional ocean currents in driving the sub-shelf melt. In addition, our ocean temperature forcing follows the evolution of the NGRIP ice core. In reality, ocean temperature variations may decouple from the atmospheric temperature variations as

reconstructed from the Greenland ice core [91]. The coupling between the ice sheet, ocean and atmosphere involves many feedbacks, such as ice shelf-ocean interaction [92–94], ice geometry and precipitation feedback, and meltwater induced ocean circulation feedback [58]. These processes should be improved by a high performance coupled earth system model in the future.

Paleoclimate proxies [7, 11, 62, 80, 82, 95–97] indicate that the Northern Hemisphere summer temperature was warmer during the Holocene than the Pre-Industrial era, primarily due to the fact that the perihelion occurred during the boreal summer. This would logically drive the retreat of the GrIS. Indeed, reconstructions of ice sheet extent [11, 98] and surface elevations [8] indicate a substantial shrinking of the GrIS during the mid-Holocene, which occurs several millennia after the Holocene thermal maximum. Our model results imply that the GrIS was not in an equilibrium state during the entire Holocene, despite a relatively stable Holocene climate. After achieving a minimum extent around 6–5 ka, the GrIS kept growing until the industrial era in response to the progressive summer cooling during the Holocene. This is in agreement with paleo reconstructions [5, 11, 16, 18].

Our simulations hint that the late Holocene growth GrIS was reversed by anthropogenic global warming after the late 20th century. Satellite observations [99] and recent reconstructions of mass balance [100] and margin-position [101] suggest no prolonged mass loss of the GrIS before the 1980s, even though the air temperature had already risen globally [102] and in Greenland [103, 104] for more than a century. This implies that during the 20th century, the GrIS may even have counteracted sea level rise due other factors, such as thermal expansion and melting glaciers. This, consequently, might mean those factors are underestimated. The slow response of the GrIS to climate change highlights the critical role of past climate change on the present and future evolution of the GrIS. Our sensitivity GrIS simulations with different initialisation strategies indicate that the onset and magnitude of the GrIS melting can both be overestimated if the background trend due to the late-Holocene GrIS growth is not taken into account.

Previously, many activities have been carried out to evaluate the impact of initialisation on the projections of ice-sheet evolution, such as the initialisation intercomparison project (init-MIP, [33, 105]). In the context of the initMIP, three major initialisation approaches are usually applied: paleo-spinup, equilibrium-spinup, and assimilation of ice sheet geometry and velocity based on observations. Due to the slow response time of the GrIS to climate change, we suggest that the GrIS was not in an equilibrium state around the Pre-Industrial era. There is very likely a background trend of growth of the GrIS approaching to the Industrial era. Therefore, the equilibrium-spinup may not be a good way to initialise the GrIS. Based on our results, we suggest that properly reproducing the ice velocity, or the rate of dynamic ice loss, is necessary to project the future evolution of the GrIS correctly. Thus, the paleo-spinup approach is proposed to be the better method to initialise the GrIS than the equilibrium-spinup. Previous sensitivity tests [23] also found that simulations with paleo-spinup produce results closer to observations than those with an equilibrium-spinup.

If the understanding of the GrIS evolution is only limited to centennial timescales, model projections indicate that the GrIS will probably contribute 5 cm to 33 cm sea level rise under the most extreme warming scenario (RCP8.5) by the end of 21th century [31, 106, 107]. This seems to indicate that the contribution of GrIS melting to sea level rise is only on the order of tens of centimetre, in the worst case. However, fossil plants have been found under the 1.4 km ice in northwestern GrIS, which were dated back to a time between the late Pliocene and early Pleistocene. At that time, the greenhouse gases level was similar or even lower than today [108, 109], hinting that the GrIS may actually be very sensitive to climate change, especially in the context of the equilibrium response. We highlight that the ice volume response of the GrIS

strongly lags climate change. Our simulations show that the warming of Greenland during the last deglaciation was reversed around 8 ka, when Greenland's summer temperature reached its peak value. However, despite cooling in the climate forcing, the GrIS still continuously lost mass for several thousand years from 8 ka to the mid-Holocene. The delayed response reminds us that significant sea level rise may occur irreversibly and last for several millennia even if the ongoing climate warming stops or reverses. Our preliminary simulation into the far future indicates that the GrIS is very likely to melt away under the RCP8.5 climate scenario, if the warming anomaly is kept constant at the 2100 levels. Long-term simulations [31] suggest that Greenland is likely to be ice free within a millennium if there is further warming after 2100 and no action to tackle the anthropogenic warming. Therefore, understanding and predicting the evolution of the GrIS must be extended for a longer period, on the timescale of millennia.

## Conclusions

By focusing on the impact of paleoclimate on the present and future evolution of the GrIS, we simulate the GrIS from the last interglacial (125 ka) to 2100 AD by using climate forcing from a comprehensive, coupled climate model. Our model results reveal that the sea level response of the GrIS lagged climate changes by several millennia. This indicates that the observed evolution of the GrIS is not only a result of present climate change, but also affected by paleoclimate, especially the transition from the relatively warm Holocene climate to cooler pre-industrial conditions. We highlight that the GrIS was very likely still growing until the late 20th century due to the background summer cooling from the mid-Holocene to the Pre-Industrial era. Without including such a background trend, the onset and magnitude of the GrIS melt can both be overestimated by an ice sheet model.

## Supporting information

**S1 File.**
(PDF)

## Acknowledgments

We would like to acknowledge Tijn Berends and another two anonymous reviewer for providing constructive comments. We would also like to thank the AWI supercomputer centre (Ollie), especially our colleagues Malte Thoma and Natalja Rakowsky, for supporting the simulations of this study.

## Author Contributions

**Conceptualization:** Thomas Kleiner.

**Funding acquisition:** Uta Krebs-Kanzow.

**Investigation:** Hu Yang, Uta Krebs-Kanzow, Dmitry Sidorenko, Christian Bernd Rodehacke, Lennert B. Stap, Gerrit Lohmann.

**Methodology:** Hu Yang, Uta Krebs-Kanzow, Thomas Kleiner, Christian Bernd Rodehacke, Xiaoxu Shi, Paul Gierz, Lu Niu, Evan J. Gowan, Sebastian Hinck, Xingxing Liu.

**Resources:** Gerrit Lohmann.

**Software:** Thomas Kleiner, Dmitry Sidorenko, Paul Gierz, Lu Niu, Sebastian Hinck, Lennert B. Stap.

**Supervision:** Uta Krebs-Kanzow, Christian Bernd Rodehacke, Gerrit Lohmann.

**Validation:** Hu Yang, Uta Krebs-Kanzow.

**Visualization:** Hu Yang, Xiaoxu Shi.

**Writing – original draft:** Hu Yang, Uta Krebs-Kanzow.

**Writing – review & editing:** Hu Yang, Uta Krebs-Kanzow, Thomas Kleiner, Dmitry Sidorenko, Christian Bernd Rodehacke, Xiaoxu Shi, Paul Gierz, Evan J. Gowan, Sebastian Hinck, Xingxing Liu, Lennert B. Stap, Gerrit Lohmann.

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
