## [Decision Letter · Decision Letter 0]

8 Jul 2021

PONE-D-21-18343

Impact of paleo climate on present and future evolution of Greenland Ice Sheet

PLOS ONE

Dear Dr. Yang,

Thank you for submitting your manuscript to PLOS ONE. After careful consideration, we feel that it has merit but does not fully meet PLOS ONE’s publication criteria as it currently stands. Therefore, we invite you to submit a revised version of the manuscript that addresses the points raised during the review process.

The revised version of the manuscript have been seen by one of the previous reviewers and one new reviewer. As you can see below both are overall happy with the manuscript, but reviewer 1 has a major concern regarding the oceanic forcing applied in your study. I largely agree with the reviewer and would like to ask you to revise the manuscript accordingly. Moreover, please answer the other (minor) comments by both reviewers.

We look forward to receiving your revised manuscript.

Kind regards,

Anna von der Heydt

Academic Editor

PLOS ONE

Journal Requirements:

2. Please update your submission to use the PLOS LaTeX template. The template and more information on our requirements for LaTeX submissions can be found at http://journals.plos.org/plosone/s/latex

[This work was supported through grant (Global sea level change since the Mid Holocene: Background trends and climate-ice sheet feedbacks) from the Deutsche Forschungsgemeinschaft (DFG) as part of the Special Priority Program (SPP)-1889 “Regional Sea Level Change and Society” (SeaLevel). C. Rodehacke has been financed through the German Federal Ministry of Education and Research (Bundesministerium f¨ur Bildung und Forschung: BMBF) project ZUWEISS 

(grant agreement 01LS1612A) and through the National Centre for Climate Research (NCFK, Nationalt Center for Klimaforskning) provided by the Danish State. H.Y. and X.L are partly funded by the open fund of State Key Laboratory of Loess and Quaternary Geology, Institute of Earth Environment, CAS (SKLLQG1920). Development of PISM is supported by NASA grant NNX17AG65G and NSF grants PLR-1603799 and PLR-1644277.]

 [This work was supported through grant (Global sea level change since the Mid

Holocene: Background trends and climate-ice sheet feedbacks) from the Deutsche

Forschungsgemeinschaft (DFG) as part of the Special Priority Program (SPP)-1889

`Regional Sea Level Change and Society' (SeaLevel). C. Rodehacke has been

financed through the German Federal Ministry of Education and Research

(Bundesministerium fur Bildung und Forschung: BMBF) project ZUWEISS (grant agreement 01LS1612A) and through the National Centre for Climate Research (NCFK, Nationalt Center for Klimaforskning) provided by the Danish State. H.Y. and X.L are partly funded by the open fund of State Key Laboratory of Loess and Quaternary Geology, Institute of Earth Environment, CAS (SKLLQG1920). Development of PISM is supported by NASA grant NNX17AG65G and NSF grants PLR-1603799 and PLR-1644277.]

Additionally, because some of your funding information pertains to [commercial funding//patents], we ask you to provide an updated Competing Interests statement, declaring all sources of commercial funding. 

In your Competing Interests statement, please confirm that your commercial funding does not alter your adherence to PLOS ONE Editorial policies and criteria by including the following statement: "This does not alter our adherence to PLOS ONE policies on sharing data and materials.” as detailed online in our guide for authors  http://journals.plos.org/plosone/s/competing-interests.  If this statement is not true and your adherence to PLOS policies on sharing data and materials is altered, please explain how. 

Please include the updated Competing Interests Statement and Funding Statement in your cover letter. We will change the online submission form on your behalf.

5. We note that Figures 1, 2, 3, 8, S1, S2 and S3 in your submission contain map images which may be copyrighted. All PLOS content is published under the Creative Commons Attribution License (CC BY 4.0), which means that the manuscript, images, and Supporting Information files will be freely available online, and any third party is permitted to access, download, copy, distribute, and use these materials in any way, even commercially, with proper attribution. For these reasons, we cannot publish previously copyrighted maps or satellite images created using proprietary data, such as Google software (Google Maps, Street View, and Earth). For more information, see our copyright guidelines: http://journals.plos.org/plosone/s/licenses-and-copyright.

a) You may seek permission from the original copyright holder of Figures Figures 1, 2, 3, 8, S1, S2 and S3 to publish the content specifically under the CC BY 4.0 license.  

6. Please ensure that you refer to Figure 7 in your text as, if accepted, production will need this reference to link the reader to the figure.

Reviewers' comments:

Reviewer's Responses to Questions

**Comments to the Author**

1. Is the manuscript technically sound, and do the data support the conclusions?

Reviewer #1: Yes

Reviewer #2: Yes

2. Has the statistical analysis been performed appropriately and rigorously? 

Reviewer #1: N/A

Reviewer #2: N/A

3. Have the authors made all data underlying the findings in their manuscript fully available?

Reviewer #1: Yes

Reviewer #2: Yes

4. Is the manuscript presented in an intelligible fashion and written in standard English?

Reviewer #1: No

Reviewer #2: Yes

5. Review Comments to the Author

Reviewer #1: In this work the authors present simulations of the Greenland Ice Sheet (GrIS) evolution from the last deglaciation to 2100 performed by means of an ice-sheet – earth system model framework coupled in an offline mode. The authors suggest that, due to the time lag in the ice sheet response to the climate forcing, the 20th century retreat still shows some reminiscence of the late Holocene cooling. This also applies to future mass-loss projections, meaning that a paleoclimate spinup is required not to overestimate the future ice-sheet contribution to sea-level (SL) rise.

The experiment conducted here and the results achieved are certainly interesting. However, I believe there are still one major concern that the authors did not completely solve after the first round of review.

As other reviewers already pointed out, the oceanic forcing implemented in this work is inappropriate, as considering constant oceanic conditions since the last deglaciation is really unrealistic. In this experimental setup, the coastal retreat is only ensured by sea level changes through the flotation criterion, while any ocean-induced melting at the ice-water interface is neglected. The lack of evidence suggesting an ice shelf as large as those of Antarctica, as the author state, does not guarantee that subshelf melting played a minor role in the retreat. In fact, it’s plenty of work showing 1. the crucial role of the ocean in the past retreat since the last deglaciation (e.g. Jennings et al., 2017, Syring et al., 2020), but also 2. the importance of modelling oceanic changes to well simulate the past (e.g. Bradley et al., 2018, Tabone et al., 2018), 3. the current state (e.g. Rignot et al., 2016, Cowton et al., 2018, Morlighem et al., 2016) and 4. the future (Slater et al., 2020, Goelzer et al., 2020; Choi et al., 2017) of the Greenland ice sheet. Experiments such as ISMIP6 (Goelzer et al., 2020), yet with simple oceanic melt parametrisations, take into account oceanic changes in future projections, although the current percentage of the GrIS in contact with the ocean will likely decrease more.

Since this study focuses on past and future retreat simulations, I would strongly suggest to consider at least a time-dependent oceanic temperature To to capture the oceanic variability at regional spatial scales. This would mean repeating the experiments, but since the authors have already expressed their unwillingness to do it, this issue should be at least discussed thoroughly as it’s a critical flaw of the research. I don’t think this is properly done in the current version of the manuscript.

Another (minor) concern is related to the climate forcing strategy applied to the last 21ka. Why not using the same climatic index applied between 128 ka to 21 ka? In this way, abrupt events such as the Bolling Allerod could be perceived by the climatic forcing and reflected in the ice-sheet retreat. Are the authors sure that a linear interpolation until the HTM does not affect the climatic background that the GrIS would be still subjected to? I just found strange that such attention to the millennial variability is reserved for the spinup and then neglected for the most interesting part of the experiment.

List of minor changes (only those I picked, please read the manuscript carefully to avoid typos/english mistakes):

Title: please, change it to “Impact of paleo climate on present and future evolution of the Greenland Ice Sheet”

Abstract: line 7, “… the climate of Greenland”.

Age: I’d rather see “26 ka BP” or “26 ka ago” or “-26 ka” instead of “26 ka”. Please, check the journal policy on that.

Author summary: line 11, “of the GrIS is not only controlled...”

Line 9 (also further in the text): extent not extend

Line 36: “Considering that the available studies focus on the evolution of the GrIS either in

the past or into the future, the impact of past climate on present and future evolution

of the GrIS is not well known”. This might be true, but the authors are completely bypassing the fact that most of work of stand-alone ice sheet models cited here does proper glacial spinups to perform future projections (e.g. Applegate et al., 2012, Yan et al., 2013, Calov et al., 2018, ... and also Fuerst et al., 2015, Rueckamp et al. 2019, not cited) thus there is a certain “paleoclimatic background” in those future simulations. Please, add a small paragraph on this, especially since it’s related to your conclusions. Moreover, I found the introduction pretty short for the work you are presenting. Please, consider adding some relevant work focusing on the past (e.g. Buizert et al., 2018, Tabone et al., 2018) and the future (e.g. Fuerst et al., 2015, Rueckamp et al. 2019) evolution of the ice sheet.

Line 38: change to “we simulate the evolution...”

Line 40 “which do not contain...”

Line 59: “It should be noted…”

Section 1.3: I suggest to precisely describe the parameters perturbed in the ensemble simulation instead of putting the PISM user’s manual reference. Which is the “SIA” parameter, for instance? Which is the basal sliding parameter considered? I think this would be much interesting for a broad audience of ice-sheet modellers than simply look at the parameter name and try to guess what it means.

Line 113: “we perform”

Line 162: “the 21 ka experiment features...”

Line 268: “model simulations reveal...”

Line 311: “the GrIS has undergone...”

References

Applegate, P. J., Kirchner, N., Stone, E. J., Keller, K., & Greve, R. (2012). An assessment of key model parametric uncertainties in projections of Greenland Ice Sheet behavior. The Cryosphere, 6(3), 589-606.

Bradley, S. L., Reerink, T. J., Van De Wal, R. S., & Helsen, M. M. (2018). Simulation of the Greenland Ice Sheet over two glacial–interglacial cycles: investigating a sub-ice-shelf melt parameterization and relative sea level forcing in an ice-sheet–ice-shelf model. Climate of the Past, 14(5), 619-635.

Buizert, C., Keisling, B. A., Box, J. E., He, F., Carlson, A. E., Sinclair, G., & DeConto, R. M. (2018). Greenland‐wide seasonal temperatures during the last deglaciation. Geophysical Research Letters, 45(4), 1905-1914.

Calov, R., Beyer, S., Greve, R., Beckmann, J., Willeit, M., Kleiner, T., ... & Ganopolski, A. (2018). Simulation of the future sea level contribution of Greenland with a new glacial system model. The Cryosphere, 12(10), 3097-3121.

Choi, Y., Morlighem, M., Rignot, E., Mouginot, J., & Wood, M. (2017). Modeling the response of Nioghalvfjerdsfjorden and Zachariae Isstrøm Glaciers, Greenland, to ocean forcing over the next century. Geophysical Research Letters, 44(21), 11-071.

Cowton, T. R., Sole, A. J., Nienow, P. W., Slater, D. A., & Christoffersen, P. (2018). Linear response of east Greenland’s tidewater glaciers to ocean/atmosphere warming. Proceedings of the National Academy of Sciences, 115(31), 7907-7912.

Fürst, J. J., Goelzer, H., & Huybrechts, P. (2015). Ice-dynamic projections of the Greenland ice sheet in response to atmospheric and oceanic warming. The Cryosphere, 9(3), 1039-1062.

Goelzer, H., Nowicki, S., Payne, A., Larour, E., Seroussi, H., Lipscomb, W. H., ... & van den Broeke, M. (2020). The future sea-level contribution of the Greenland ice sheet: a multi-model ensemble study of ISMIP6. The Cryosphere, 14(9), 3071-3096.

Jennings, A. E., Andrews, J. T., Cofaigh, C. Ó., Onge, G. S., Sheldon, C., Belt, S. T., ... & Hillaire-Marcel, C. (2017). Ocean forcing of Ice Sheet retreat in central west Greenland from LGM to the early Holocene. Earth and Planetary Science Letters, 472, 1-13.

Morlighem, M., Bondzio, J., Seroussi, H., Rignot, E., Larour, E., Humbert, A., & Rebuffi, S. (2016). Modeling of Store Gletscher's calving dynamics, West Greenland, in response to ocean thermal forcing. Geophysical Research Letters, 43(6), 2659-2666.

Rignot, E., Xu, Y., Menemenlis, D., Mouginot, J., Scheuchl, B., Li, X., ... & Fleurian, B. D. (2016). Modeling of ocean‐induced ice melt rates of five west Greenland glaciers over the past two decades. Geophysical Research Letters, 43(12), 6374-6382.

Rückamp, M., Greve, R., & Humbert, A. (2019). Comparative simulations of the evolution of the Greenland ice sheet under simplified Paris Agreement scenarios with the models SICOPOLIS and ISSM. Polar Science, 21, 14-25.

Syring, N., Lloyd, J. M., Stein, R., Fahl, K., Roberts, D. H., Callard, L., & O'Cofaigh, C. (2020). Holocene Interactions Between Glacier Retreat, Sea Ice Formation, and Atlantic Water Advection at the Inner Northeast Greenland Continental Shelf. Paleoceanography and Paleoclimatology, 35(11), e2020PA004019.

Slater, D. A., Felikson, D., Straneo, F., Goelzer, H., Little, C. M., Morlighem, M., ... & Nowicki, S. (2020). Twenty-first century ocean forcing of the Greenland ice sheet for modelling of sea level contribution. The Cryosphere, 14(3), 985-1008.

Tabone, I., Blasco, J., Robinson, A., Alvarez-Solas, J., & Montoya, M. (2018). The sensitivity of the Greenland Ice Sheet to glacial–interglacial oceanic forcing. Climate of the Past, 14(4), 455-472.

Yan, Q., Zhang, Z., Gao, Y., Wang, H., & Johannessen, O. M. (2013). Sensitivity of the modeled present‐day Greenland Ice Sheet to climatic forcing and spin‐up methods and its influence on future sea level projections. Journal of Geophysical Research: Earth Surface, 118(4), 2174-2189.

Reviewer #2: Review of “Impact of paleo climate on present and future evolution of Greenland Ice Sheet” by Tijn Berends

General comments

The authors present a revised version of a manuscript describing simulations of the Greenland ice sheet throughout the last glacial cycle and into the near future, using the ice-sheet model PISM forced with output of the AWI-ESM climate model. They find that, in agreement with the general consensus, changes in ice volume lag changes in climate by up to several millennia. Given that the Earth’s climate, particularly in Greenland, is thought to have been relatively warm during the early Holocene, and to have cooled to pre-industrial levels during the last few thousand years, this means that the Greenland ice sheet was likely not in equilibrium, but rather was still advancing when anthropogenic climate change commenced. The authors show that this likely delayed the onset of ice-sheet retreat until several decades after anthropogenic warming started, and is even now leading to slower ice-sheet retreat than would have been the case if the Holocene climate had been completely stable. These are important findings, as they highlight the need to account for the paleoclimatic history of the ice-sheet when projecting its future evolution.

In response to comments raised in the first review round, the authors significantly altered the manuscript. They included results from several additional experiments intended to investigate the sensitivity of their results to different model parameters, which support their conclusion that the observed lag in ice volume change with respect to climate is robust. They also altered the general story of the manuscript to focus more on the impact of (recent) paleoclimate change on near-future ice-sheet evolution, which has improved the readability of the manuscript. Lastly, they adequately addressed my major concern from the first review, which was unfortunately based on a misunderstanding on my part. I think the manuscript is now fit to be published after some minor revisions.

Specific comments

Line 35: “Considering that the available studies focus on the evolution of the GrIS either in the past or into the future… ” This is phrased too strongly. Particularly the recent initMIP-GRL paper by Goelzer et al. (2018) presents a detailed study of the effect of model initialization (including paleo spin-up) on future projections. Mention this.

Line 43 (what happened to the line numbers here): “we select the Lingle and Clark bed deformation model…” I now understand the difference between your prescribed sea-level forcing and the dynamically calculated bed deformation. My previous criticism was based on a misunderstanding of this on my part, for which I apologise. I agree that the eustatic signal plus local bedrock deformation is probably sufficient for Greenland.

Line 43 (idem): “A constant temperature lapse rate of 5 ◦C km−1 is adopted…” What is this based on? It seems a little low to me (between 6 – 8 K/km is common I think).

Line 43 (idem): “The eigencalving and thickness calving parametrizations are used to determine the calving-front dynamics…” These are different approaches. Do you mean you use both of them together?

Line 57: “We tune the enhancement factor…” This requires more explanation. Do you do this in a steady-state set-up? Or do you tune it to give good results at the end of your LGC run? If so, do you also tune for LGM volume/extent in any way?

Line 60: “…we do not use the elevation induced precipitation downscaling, because…[] …our climate model has a higher spatial resolution…” Is this spatial resolution smaller than the change in ice margin position? Particularly in the south-west, where the ice margin migrates onto the continental shelf during the LGM (presumably, actually a figure showing a few time-slices of your modelled ice-sheet geometry would be very nice!), you’d expect the high-precipitation area to migrate along with it. If you don’t correct for this in some way, this might introduce a bias in your mass balance.

Line 76: “…following the protocol of the Paleoclimate Modelling 76 Intercomparison Project phase 4…” Did AWI-ESM participate in PMIP4? How do your paleoclimate time slices compare?

Line 94: “From 21ka to Pre-Industrial era, the AWI-ESM 94 time-slice simulations … are linearly 95 interpolated to derive a continuous climate forcing” Looking at Fig. 4A, this seems to introduce a warm bias during the deglaciation. Do you think this might affect your results?

Table 1: I think you changed the SIA flow enhancement factor? Why not also the SSA factor?

Section 2 in general: maybe consider summarizing this. Your article seems to be aimed mostly at ice-sheet modellers such as myself. I don’t really know enough about the details of paleoclimate at specific points in time to know what to make of this paragraph. Really all I need to know is that your paleoclimate is good. If AWI-ESM participated in PMIP4/PMIP5 (while you refer to the PMIP protocol earlier, you don’t mention if AWI-ESM participated or not) then just mentioning that would already be enough.

Line 191: “…suggesting that the surface mass balance is the primary driver for the GrIS.” Given that at present, the GrIS loses most of its mass through glacier discharge (e.g. King et a., 2018; Fettweis et al., 2020) rather than runoff, I’d phrase this differently.

Line 197: “…with an ice volume of 0.24 m SLE smaller than the Pre-Industrial volume.” I’d love to see a figure of this minimum extent.

Line 232: “…In this simulation, the initial condition is obtained by continuing the paleo GrIS simulation for ten thousand years.” In my opinion, the difference between this simulation and the “default” run is your most important result; this illustrates very clearly why including a paleo-spin-up in future projections is so important. I’d mention this one already in the introduction, and consider rearranging the text of your results to present it accordingly (as in, you’ve done your default run and this one, plus a small ensemble of model parameter sensitivity runs for both experiments, so group them like that).

Line 268: “…reveals that the 268 GrIS may retreat…” Mind your grammar.

Line 281: “Our simulated magnitudes show discrepancies with the previous studies, likely because the simulations have not been constrained by any observation.” Elaborate on this. Which previous studies are these? What ice volumes did they find?

Line 334: “Based on our results, we suggest that properly reproducing the ice velocity, or the rate of dynamic ice loss, is necessary to project the future evolution of the GrIS correctly. Thus, the paleo-spinup and assimilation approaches are proposed to be better methods to initialise the GrIS.” You’ve not done anything with data assimilation in your study, so this statement does not belong here. Whether or not a data-assimilated model will implicitly contain the information from the paleo-history of the ice-sheet is a question far beyond the scope of your study.

Line 345: “At that time the greenhouse gases level was similar or even lower than today…” CO2 proxies that far back have uncertainties of over 50 ppmv in either direction, and data points for the Late Pliocene / Early Pleistocene cover pretty much everything from 250 to 450 ppmv (see e.g. Berends et al. 2021, Clim. Past.). Also, you’re now comparing centennial-scale sea-level rise to equilibrated ice volumes – which, as is pretty much the entire point of your study, are not at all the same.

Throughout the manuscript: mind the difference in spelling between the verb “to extend” and the noun “extent”.

6. PLOS authors have the option to publish the peer review history of their article (what does this mean?). If published, this will include your full peer review and any attached files.

Reviewer #1: No

Reviewer #2: **Yes: **Tijn Berends

---

## [Author Response · Author response to Decision Letter 0]

18 Aug 2021

All the response can be found in the attached response letter

---

## [Decision Letter · Decision Letter 1]

17 Sep 2021

PONE-D-21-18343R1Impact of paleoclimate on present and future evolution of the Greenland Ice SheetPLOS ONE

Dear Dr. Yang,

Thank you for submitting your manuscript to PLOS ONE. After careful consideration, we feel that it has merit but does not fully meet PLOS ONE’s publication criteria as it currently stands. Therefore, we invite you to submit a revised version of the manuscript that addresses the points raised during the review process.

One of the reviewers has seen the revision again and finds the new version adequately addresses the raised issues. The reviewer has only minor suggestions left. Please address those in a new version of the manuscript.

We look forward to receiving your revised manuscript.

Kind regards,

Anna von der Heydt

Academic Editor

PLOS ONE

Journal Requirements:

Additional Editor Comments:

One of the reviewers has seen the revision again and finds the new version adequately addresses the raised issues. The reviewer has only minor suggestions left. Please address those in a new version of the manuscript.

Reviewers' comments:

Reviewer's Responses to Questions

**Comments to the Author**

1. If the authors have adequately addressed your comments raised in a previous round of review and you feel that this manuscript is now acceptable for publication, you may indicate that here to bypass the “Comments to the Author” section, enter your conflict of interest statement in the “Confidential to Editor” section, and submit your "Accept" recommendation.

Reviewer #1: (No Response)

2. Is the manuscript technically sound, and do the data support the conclusions?

Reviewer #1: Yes

3. Has the statistical analysis been performed appropriately and rigorously? 

Reviewer #1: N/A

4. Have the authors made all data underlying the findings in their manuscript fully available?

Reviewer #1: Yes

5. Is the manuscript presented in an intelligible fashion and written in standard English?

Reviewer #1: Yes

6. Review Comments to the Author

Reviewer #1: Yang et al. investigate the influence of the past climate of the Greenland Ice Sheet (GrIS) on its future projections of mass loss. By carrying out ice-sheet-earth-system models simulations from the last glacial period to the near future (2100 AD), they suggest that a classic long paleoclimatic spinup is needed to prevent an overestimation of predictions of Greenland ice loss.

I am glad the authors decided to redo the simulations to include the ocean as an additional forcing to the ice sheet model. I appreciate the authors' effort in this sense as I understand how this might be a considerable investment of resources, especially when dealing with paleoclimatic runs.

As the authors show in the revised manuscript, the inclusion of a spatially-uniform transient oceanic temperature forcing does not have a significant effect on the simulated GrIS evolution. I agree that this might be related to the simplistic approach adopted here. In fact, subshelf temperatures at depths lower than 100 m are likely more suitable to be considered, as closer to the ice shelf base. At those depths, the ocean water is decoupled from the surface air temperature, thus a transient oceanic temperature that follows the NGRIP time series is probably not a good representation of the paleoclimatic water temperature trend. An example can be found in southeast Greenland where warmer subsurface currents during the Younger Dryas, in clear contrast with the simultaneous atmospheric cooling, led to a significant coastal mass loss (Rainsley et al., 2018). Moreover, ocean temperatures around Greenland are strongly affected by local currents which, of course, cannot be captured if their spatial average is taken. However, since increased atmospheric temperature driven by insolation changes is likely the main responsible for the GrIS retreat during the Holocene (Vasskog et al., 2015), I would not expect that a more sophisticated approach regarding the ocean would drastically change the main results of this work, especially regarding the latest part of the Holocene. I therefore thank the authors for their effort made in satisfying my requests and I suggest the publication of the manuscript after minor revisions.

Specific comments

Abstract, line 8: please change to "maximum and minimum ice volume".

Author summary

- line 3: please change to "it remains uncertain on how fast and how much the GrIS will contribute to it".

- line 8: I don't understand this sentence very well: "These results are consistent with evidence of the times of both past extremes in climates...". Please rephrase.

- line 14: please change "growing" to "growth".

Main text:

Line 39: The short term response of ice sheet models to what? Please explain.

Line 64: This is imprecise as the PDD is an ablation scheme only. Please, correct.

Line 71: Please change to "at the ice front".

Line 75: Please change to "continental shelf of Greenland". This error repeats several times. Choose either "of Greenland" or "of the Greenland Ice Sheet".

Line 101: please change to "coupled with the".

Line 104: I think it is sufficient to write the acronym.

Line 126: please change to "the index to simulate".

Line 139: Which "situation"? Please consider changing it to "climate snapshots available before 21 ka" or something alike.

Line 143-145. I would be careful with this sentence. The parametrisation you are using (Beckmann & Goose, 2008) is built for the ice shelf base as no lateral fracturing is considered. Grounding lines in Greenland are usually deeper in the water column (Wilson et al., 2017), which contrasts your choice of considering an oceanic temperature at 100 m. I would add a brief sentence in the discussion regarding this and the possible influence that this might have on your results (see my general comments).

Line 149: I guess you wanted to put "PISM" instead of "PIK".

Line 162: I would not call them "ensemble simulations" as they are rather sensitivity runs. This is misleading as it may be understood as five sets of simulations, while here we have only five runs. I would prefer to read it as "five simulations" or "five sensitivity tests" but I leave it to the authors' discretion.

Line 176: "one more ensemble simulation" as above.

Line 195: what do you mean with "there are spatially heterogeneous anomalies at 9 ka"? Do you want to point out that the west Greenland is cooler than the Pre-industrial level, while the rest of the ice sheet is warmer? Please explain.

Line 197: "This implies a long term cooling trend ...". The sentence before does not allow to get to this conclusion. Please, rephrase.

Line 214: Figure S1 maybe?

Line 219: Again, Fig. 6 is likely wrong.

Line 220: Was the acronym LGM presented before in the text?

Line 228: "is relatively heterogeneous" maybe?

Line 241: Please change to "the minimum ice volume".

Line 249: Please check the figure numbers.

Line 282-284: Figure 2?

Line 298-300: I don't understand why the northeast has a higher volume if the precipitation rate is lower. Could you explain it better?

Line 310: is this ice surface temperature or is it averaged over the ice column?

Line 312: Would you mean "the ice temperature of the GrIS is still increasing ..."?

Line 406-415: This paragraph needs some rephrasing in my view. Contributing to 33mm to sea level rise by 2100 does not imply that the GrIS is rather little sensitive to climate change. This is a subjective point of view.

References:

Beckmann, A., & Goosse, H. (2003). A parameterization of ice shelf–ocean interaction for climate models. Ocean modelling, 5(2), 157-170.

Rainsley, E., Menviel, L., Fogwill, C. J., Turney, C. S., Hughes, A. L., & Rood, D. H. (2018). Greenland ice mass loss during the Younger Dryas driven by Atlantic Meridional Overturning Circulation feedbacks. Scientific reports, 8(1), 1-9.

Vasskog, K., Langebroek, P. M., Andrews, J. T., Nilsen, J. E. Ø., & Nesje, A. (2015). The Greenland Ice Sheet during the last glacial cycle: Current ice loss and contribution to sea-level rise from a palaeoclimatic perspective. Earth-Science Reviews, 150, 45-67.

Wilson, N., Straneo, F., & Heimbach, P. (2017). Satellite-derived submarine melt rates and mass balance (2011–2015) for Greenland's largest remaining ice tongues. The Cryosphere, 11(6), 2773-2782.

7. PLOS authors have the option to publish the peer review history of their article (what does this mean?). If published, this will include your full peer review and any attached files.

Reviewer #1: No

---

## [Author Response · Author response to Decision Letter 1]

8 Oct 2021

All response are included in the response letter.

---

## [Editor Report · Decision Letter 2]

27 Oct 2021

Impact of paleoclimate on present and future evolution of the Greenland Ice Sheet

PONE-D-21-18343R2

Dear Dr. Yang,

We’re pleased to inform you that your manuscript has been judged scientifically suitable for publication and will be formally accepted for publication once it meets all outstanding technical requirements.

Kind regards,

Anna von der Heydt

Academic Editor

PLOS ONE

Additional Editor Comments (optional):

Thank you for submitting a revised version. I can see that you have addressed all remaining issues of the reviewer and I am happy to accept the manuscript for publication in PLOS One
---

## [Editor Report · Acceptance letter]

29 Oct 2021

PONE-D-21-18343R2 

Impact of paleoclimate on present and future evolution of the Greenland Ice Sheet  

Dear Dr. Yang:

I'm pleased to inform you that your manuscript has been deemed suitable for publication in PLOS ONE. Congratulations! Your manuscript is now with our production department. 

Kind regards, 

on behalf of

Dr. Anna von der Heydt 

Academic Editor

PLOS ONE